# Sample Complexity of Learning Mixtures of Sparse Linear Regressions

**Akshay Krishnamurthy**
Microsoft Research, NYC
akshay@cs.umass.edu

**Arya Mazumdar**
UMass Amherst
arya@cs.umass.edu

**Andrew McGregor**
UMass Amherst
mcgregor@cs.umass.edu

**Soumyabrata Pal**
UMass Amherst
spal@cs.umass.edu

## Abstract

In the problem of *learning mixtures of linear regressions*, the goal is to learn a collection of signal vectors from a sequence of (possibly noisy) linear measurements, where each measurement is evaluated on an unknown signal drawn uniformly from this collection. This setting is quite expressive and has been studied both in terms of practical applications and for the sake of establishing theoretical guarantees. In this paper, we consider the case where the signal vectors are *sparse*; this generalizes the popular compressed sensing paradigm. We improve upon the state-of-the-art results as follows: In the noisy case, we resolve an open question of Yin et al. (IEEE Transactions on Information Theory, 2019) by showing how to handle collections of more than two vectors and present the first robust reconstruction algorithm, i.e., if the signals are not perfectly sparse, we still learn a good sparse approximation of the signals. In the noiseless case, as well as in the noisy case, we show how to circumvent the need for a restrictive assumption required in the previous work. Our techniques are quite different from those in the previous work: for the noiseless case, we rely on a property of sparse polynomials and for the noisy case, we provide new connections to learning Gaussian mixtures and use ideas from the theory of error correcting codes.

## 1   Introduction

Learning mixtures of linear regressions is a natural generalization of the basic linear regression problem. In the basic problem, the goal is to learn the best linear relationship between the scalar responses (i.e., labels) and the explanatory variables (i.e., features). In the generalization, each scalar response is stochastically generated by picking a function uniformly from a set of $L$ unknown linear functions, evaluating this function on the explanatory variables and possibly adding noise; the goal is to learn the set of $L$ unknown linear functions. The problem was introduced by De Veaux [11] over thirty years ago and has recently attracted growing interest [8, 14, 22, 24, 25, 27]. Recent work focuses on a query-based scenario in which the input to the randomly chosen linear function can be specified by the learner. The *sparse* setting, in which each linear function depends on only a small number of variables, was recently considered by Yin et al. [27], and can be viewed as a generalization of the well-studied compressed sensing problem [7, 13]. The problem has numerous applications in modelling heterogeneous data arising in medical applications, behavioral health, and music perception [27].

**Formal Problem Statement.** There are $L$ unknown distinct vectors $\beta^1, \beta^2, \ldots, \beta^L \in \mathbb{R}^n$ and each is $k$-sparse, i.e., the number of non-zero entries in each $\beta^i$ is at most $k$ where $k$ is some known parameter. We define an oracle $\mathcal{O}$ which, when queried with a vector $\mathbf{x} \in \mathbb{R}^n$, returns the noisy output $y \in \mathbb{R}$:

$$y = \langle \mathbf{x}, \boldsymbol{\beta} \rangle + \eta \tag{1}$$

where $\eta$ is a random variable with $\mathbb{E}\eta = 0$ that represents the measurement noise and $\boldsymbol{\beta}$ is chosen uniformly[1] from the set $\mathcal{B} = \{\beta^1, \beta^2, \ldots, \beta^L\}$. The goal is to recover all vectors in $\mathcal{B}$ by making a set of queries $\mathbf{x}_1, \mathbf{x}_2, \ldots, \mathbf{x}_m$ to the oracle. We refer to the values returned by the oracle given these queries as *samples*. Note that the case of $L = 1$ corresponds to the problem of compressed sensing. Our primary focus is on the sample complexity of the problem, i.e., minimizing the number of queries that suffices to recover the sparse vectors up to some tolerable error.

**Related Work.** The most relevant previous work is by Yin et al. [27]. For the noiseless case, i.e., $\eta = 0$, they show that $O(kL \log(kL))$ queries are sufficient to recover all vectors in $\mathcal{B}$ with high probability. However, their result requires a restrictive assumption on the set of vectors and do not hold for an arbitrary set of sparse vectors. Specifically, they require that for any $\beta, \beta' \in \mathcal{B}$,

$$\beta_j \neq \beta'_j \quad \text{for each} \quad j \in \operatorname{supp}(\beta) \cap \operatorname{supp}(\beta') . \tag{2}$$

Their approach depends crucially on this assumption and this limits the applicability of their approach. Note that our results will not depend on such an assumption. For the noisy case, the approach taken by Yin et al. only handles the $L = 2$ case and they state the case of $L > 2$ as an important open problem. Resolving this open problem will be another one of our contributions.

More generally, both compressed sensing [7, 13] and learning mixtures of distributions [10, 23] are immensely popular topics across statistics, signal processing and machine learning with a large body of prior work. Mixture of linear regressions is a natural synthesis of mixture models and linear regression, a very basic machine learning primitive [11]. Most of the work on the problem has considered learning generic vectors, i.e., not necessary sparse, and they propose a variety of algorithmic techniques to obtain polynomial sample complexity [8, 14, 19, 24, 26]. To the best of our knowledge, Städler et al. [22] were the first to impose sparsity on the solutions. However, many of the earlier papers on mixtures of linear regression, essentially consider the queries to be fixed, i.e., part of the input, whereas in this paper, and in Yin et al. [27], we are interested in designing queries in such a way to minimize the number of queries.

**Our Results and Techniques.** We present results for both the noiseless and noisy cases. The latter is significantly more involved and is the main technical contribution of this paper.

*Noiseless Case:* In the case where there is no noise and the $L$ unknown vectors are $k$-sparse, we show that $O(kL \log(kL))$ queries suffice and that $\Omega(kL)$ queries are necessary. The upper bound matches the query complexity of the result by Yin et al. but our result applies for all $k$-sparse vectors rather than just those satisfying the assumption in Eq. 2. The approach we take is as follows: In compressed sensing, exact recovery of $k$-sparse vectors is possible by taking samples with an $m \times n$ matrix with any $2k$ columns linearly independent. Such matrices exists with $m = 2k$ (such as Vandermonde matrices) and are called MDS matrices. We use rows of such a matrix repeatedly to generate samples. Since there are $L$ different vectors in the mixture, with $O(L \log L)$ measurements with a row we will be able to see the samples corresponding to each of the $L$ vectors with that row. However, even if this is true for measurements with each rows, we will still not be able to align measurements across the rows. For example, even though we will obtain $\langle \mathbf{x}, \beta^\ell \rangle$ for all $\ell \in [L]$ and for all $\mathbf{x}$ that are rows of an MDS matrix, we will be unable to identify the samples corresponding to $\beta^1$. To tackle this problem, we propose using a special type of MDS matrix that allows us to align measurements corresponding to the same $\beta$s. After that, we just use the sparse recovery property of the MDS matrix to individually recover each of the vectors.

*Noisy Case:* We assume that the noise $\eta$ is a Gaussian random variable with zero mean. Going forward, we write $\mathcal{N}(\mu, \sigma^2)$ to denote a Gaussian distribution with mean $\mu$ and variance $\sigma^2$. Furthermore, we will no longer assume vectors in $\mathcal{B}$ are necessarily sparse. From the noisy samples, our objective is to

recover an estimate $\hat{\beta}$ for each $\beta \in \mathcal{B}$ such that

$$\|\beta - \hat{\beta}\| \le c\|\beta - \beta^*\|, \tag{3}$$

where $c$ is an absolute constant and $\beta^*$ is the best $k$-sparse approximation of $\beta$, i.e., all except the largest (by absolute value) $k$ coordinates set to $0$. The norms in the above equation can be arbitrary defining the strength of the guarantee, e.g., when we refer to an $\ell_1/\ell_1$ guarantee both norms are $\|\cdot\|_1$. Our results should be contrasted with [27], where results not only hold for only $L = 2$ and under assumption (2), but the vectors are also strictly $k$-sparse. However, like [27], we assume $\epsilon$-precision of the unknown vectors, i.e., the value in each coordinate of each $\beta \in \mathcal{B}$ is an integer multiple of $\epsilon$.[2]

Notice that in this model the noise is additive and not multiplicative. Hence, it is possible to increase the $\ell_2$ norm of the queries arbitrarily so that the noise becomes inconsequential. However, in a real setting, this cannot be allowed since increasing the strength (norm) of the queries has a cost and it is in our interest to minimize the cost. Suppose the algorithm designs the $i^{th}$ query vector by first choosing a distribution $Q_i$ and subsequently sampling a query vector $\mathbf{x}_i \sim Q_i$. Let us now define the signal to noise ratio as follows:

$$\mathsf{SNR} = \max_i \min_\ell \frac{\mathbb{E}_{\mathbf{x}_i \sim Q_i} |\langle \mathbf{x}_i, \beta^\ell \rangle|^2}{\mathbb{E}\eta^2} . \tag{4}$$

Our objective in the noisy setting is to recover the unknown vectors $\beta^1, \beta^2, \ldots, \beta^L \in \mathbb{R}^n$ while minimizing the number of queries and the $\mathsf{SNR}$ at the same time. In this setting, assuming that all the unknown vectors have unit norm, we show that $O(k \log^3 n \exp((\sigma/\epsilon)^{2/3}))$ queries with $\mathsf{SNR} = O(1/\sigma^2)$ suffice to reconstruct the $L = O(1)$ vectors in $\mathcal{B}$ with the approximation guarantees given in Eq. (3) with high probability if the noise $\eta$ is a zero mean gaussian with a variance of $\sigma^2$. This is equivalent to stating that $O(k \log^3 n \exp(1/(\epsilon\sqrt{\mathsf{SNR}})^{2/3}))$ queries suffice to recover the $L$ unknown vectors with high probability.

Note that in the previous work $\epsilon\sqrt{\mathsf{SNR}}$ is assumed to be at least constant and, if this is the case, our result is optimal up to polynomial factors since $\Omega(k)$ queries are required even if $L = 1$. More generally, the dependence upon $\epsilon\sqrt{\mathsf{SNR}}$ in our result improves upon the dependence in the result by Yin et al. Note that we assumed $L = O(1)$ in our result because the dependence of sample complexity on $L$ is complicated as it is implicit in the signal-to-noise ratio.

As in noiseless case, our approach is to use a compressed sensing matrix and use its rows multiple time as queries to the oracle. At the first step, we would like to separate out the different $\beta$s from their samples with the same rows. Unlike the noiseless case, even this turns out to be a difficult task. Under the assumption of Gaussian noise, however, we are able to show that this is equivalent to learning a mixture of Gaussians with different means. In this case, the means of the Gaussians belong to an "$\epsilon$-grid", because of the assumption on the precision of $\beta$s. This is not a standard setting in the literature of learning Gaussian mixtures, e.g., [1, 16, 20]. Note that, this is possible if the vector that we are sampling with has integer entries. As we will see a binary-valued compressed sensing matrix will do the job for us. We will rely on a novel complex-analytic technique to exactly learn the means of a mixture of Gaussians, with means belonging to an $\epsilon$-grid. This technique is paralleled by the recent developments in trace reconstructions where similar methods were used for learning a mixture of binomials [18, 21].

Once for each query, the samples are separated, we are still tasked with aligning them so that we know the samples produced by the same $\beta$ across different queries. The method for the noiseless case fails to work here. Instead, we use a new method motivated by error-correcting codes. In particular, we perform several redundant queries, that help us to do this alignment. For example, in addition to the pair of queries $\mathbf{x}_i, \mathbf{x}_j$, we also perform the queries defined by $\mathbf{x}_i + \mathbf{x}_j$ and $\mathbf{x}_i - \mathbf{x}_j$.

After the alignment, we use the compressed sensing recovery to estimate the unknown vectors. For this, we must start with a matrix that with minimal number of rows, will allow us to recover any vector with a guarantee such as (3). On top of this, we also need the matrix to have integer entries so that we can use our method of learning a mixture of Gaussians with means on an $\epsilon$-grid. Fortunately, a random binary $\pm 1$ matrix satisfies all the requirements [3]. Putting now these three steps of learning mixtures, aligning and compressed sensing, lets us arrive at our results.

While we concentrate on sample complexity in this paper, our algorithm for the noiseless case is computationally efficient, and the only computationally inefficient step in the general noisy case is

**Algorithm 1** `Noiseless Recovery` The algorithm for extracting recovering vectors via queries to oracle in noiseless setting.

---
**Require:** Number of unknown sparse vectors $L$, dimension $n$, sparsity $k$.
 1: Let $t \in_R \{0, 1, 2, \ldots, k^2 L^2 - 1\}$ and define $\alpha_1, \alpha_2, \ldots, \alpha_{2k}$ where $\alpha_j = \frac{2kt+j}{2k^3 L^2}$.
 2: **for** $i = 1, 2, \ldots, 2k$ **do**
 3:    Make $L \log(Lk^2)$ oracle queries with vector $[1 \ \alpha_i \ \alpha_i^2 \ \ldots \ \alpha_i^{n-1}]$. Refer to these as a *batch*.
 4: **end for**
 5: **for** $i = 1, 2, \ldots, 2k$ **do**
 6:    For each batch of query responses corresponding to the same query vector, retain unique values and sort them in ascending order. Refer to this as the *processed batch*.
 7: **end for**
 8: Set matrix $\mathcal{Q}$ of dimension $2k \times L$ such that its $j^{th}$ row is the processed batch corresponding to the query vector $[1 \ \alpha_j \ \alpha_j^2 \ \ldots \ \alpha_j^{n-1}]$
 9: **for** $i = 1, 2, \ldots, L$ **do**
10:    Decode the $i^{th}$ column of the matrix $\mathcal{Q}$ to recover $\beta^i$.
11: **end for**
12: Return $\beta^1, \beta^2, \ldots, \beta^L$.

---

that of learning Gaussian mixtures. However, in practice one can perform a simple clustering (such as Lloyd's algorithm) to learn the means of the mixture.

**Organization and Notation.** In Section 2, we present our results for the noiseless case. In Section 3.1 we consider the case with noise when $L = 2$ and then consider noise and general $L$ in Section 3.2. Most proofs are deferred to the appendix in the supplementary material. Throughout, we write $x \in_R X$ to denote taking an element $x$ from a finite set $X$ uniformly at random. For $n \in \mathbb{N}$, let $[n] := \{1, 2, \ldots, n\}$.

## 2 Exact sparse vectors and noiseless samples

To begin, we deal with the case of uniform mixture of exact sparse vectors with the oracle returning noiseless answers when queried with a vector. For this case, our scheme is provided in Algorithm 1. The main result for this section is the following.

**Theorem 1.** *For a collection of $L$ vectors $\beta^1, \beta^2, \ldots, \beta^L \in \mathbb{R}^n$ such that $\|\beta^i\|_0 \leq k \ \forall i \in [L]$, one can recover all of them exactly with probability at least $1 - 3/k$ with a total of $2kL \log Lk^2$ oracle queries. See Algorithm 1.*

A Vandermonde matrix is a matrix such that the entries in each row of the matrix are in geometric progression i.e., for an $m \times n$ dimensional Vandermonde matrix the entry in the $(i, j)$th entry is $\alpha_i^j$ where $\alpha_1, \alpha_2, \ldots, \alpha_m \in \mathbb{R}$ are distinct values. We will use the following useful property of the Vandermonde matrices; see, e.g., [15, Section XIII.8] for the proof.

**Lemma 1.** *The rank of any $m \times m$ square submatrix of a Vandermonde matrix is $m$ assuming $\alpha_1, \alpha_2, \ldots, \alpha_m$ are distinct and positive.*

This implies that, with the samples from a $2k \times n$ Vandermonde matrix, a $k$-sparse vector can be exactly recovered. This is because for any two unknown vectors $\beta$ and $\hat{\beta}$, the same set of responses for all the $2k$ rows of the Vandermonde matrix implies that a $2k \times 2k$ square submatrix of the Vandermonde matrix is not full rank which is a contradiction to Lemma 1.

We are now ready to prove Theorem 1.

*Proof.* For the case of $L = 1$, note that the setting is the same as the well-known compressed sensing problem. Furthermore, suppose a $2k \times n$ matrix has the property that any $2k \times 2k$ submatrix is full rank, then using the rows of this matrix as queries is sufficient to recover any $k$-sparse vector. By Lemma 1, any $2k \times n$ Vandemonde matrix has the necessary property.

Let $\beta^1, \beta^2, \ldots, \beta^L$ be the set of unknown $k$-sparse vectors. Notice that a particular row of the Vandermonde matrix looks like $[1 \ z \ z^2 \ z^3 \ \ldots \ z^{n-1}]$ for some value of $z \in \mathbb{R}$. Therefore, for

some vector $\beta^i$ and a particular row of the Vandermonde matrix, the inner product of the two can be interpreted as a degree $n$ polynomial evaluated at $z$ such that the coefficients of the polynomial form the vector $\beta^i$. More formally, the inner product can be written as $f^i(z) = \sum_{j=0}^{n-1} \beta_j^i z^j$ where $f^i$ is the polynomial corresponding to the vector $\beta^i$. For any value $z \in \mathbb{R}^n$, we can define an ordering over the $L$ polynomials $f^1, f^2, \ldots, f^L$ such that $f^i > f^j$ iff $f^i(z) > f^j(z)$.

For two distinct indices $i, j \in [L]$, we will call the polynomial $f^i - f^j$ a *difference polynomial*. Each difference polynomial has at most $2k$ non-zero coefficients and therefore has at most $2k$ positive roots by Descartes' Rule of Signs [9]. Since there are at most $L(L-1)/2$ distinct difference polynomials, the total number of distinct values that are roots of at least one difference polynomial is less than $kL^2$. Note that if an interval does not include any of these roots, then the ordering of $f^1, \ldots, f^L$ remains consistent for any point in that interval. In particular, consider the intervals $(0, \gamma], (\gamma, 2\gamma], \ldots, (1 - \gamma, 1]$ where $\gamma = 1/(k^2 L^2)$. At most $kL^2$ of these intervals include a root of a difference polynomial and hence if we pick a random interval then with probability at least $1 - 1/k$, the ordering of $f^1, \ldots, f^L$ are consistent throughout the interval. If the interval chosen is $(t\gamma, (t+1)\gamma]$ then set $\alpha_j = t\gamma + j\gamma/(2k)$ for $j = 1, \ldots, 2k$.

Now for each value of $\alpha_i$, define the vector $\mathbf{x}_i \equiv [1 \ \alpha_i \ \alpha_i^2 \ \alpha_i^3 \ \ldots \ \alpha_i^{n-1}]$. For each $i \in [2k]$, the vector $\mathbf{x}_i$ will be used as query to the oracle repeatedly for $L \log Lk^2$ times. We will call the set of query responses from the oracle for a fixed query vector $\mathbf{x}_i$ a *batch*. For a fixed batch and $\beta^j$,

$$\Pr(\beta^j \text{ is not sampled by the oracle in the batch}) \leq \left(1 - \frac{1}{L}\right)^{L \log Lk^2} \leq e^{-\log Lk^2} = \frac{1}{Lk^2}.$$

Taking a union bound over all the vectors ($L$ of them) and all the batches ($2k$ of them), we get that in every batch every vector $\beta^j$ for $j \in [L]$ is sampled with probability at least $1 - 2/k$. Now, for each batch, we will retain the unique values (there should be exactly $L$ of them with high probability) and sort the values in each batch. Since the ordering of the polynomial remains same, after sorting, all the values in a particular position in each batch correspond to the same vector $\beta^j$ for some unknown index $j \in [L]$. We can aggregate the query responses of all the batches in each position and since there are $2k$ linear measurements corresponding to the same vector, we can recover all the unknown vectors $\beta^j$ using Lemma 1. The failure probability of this algorithm is at most $3/k$. $\qquad\square$

The following theorem establishes that our method is almost optimal in terms of sample complexity.

**Theorem 2.** *At least $2Lk$ oracle queries are necessary to recover an arbitrary set of $L$ vectors that are $k$-sparse.*

## 3 Noisy Samples and Sparse Approximation

We now consider the more general setting where the oracle is noisy and the vectors $\beta^1, \ldots, \beta^L$ are not necessarily sparse. We assume $L$ is an arbitrary constant, i.e., it does not grow with $n$ or $k$ and that the unknown vectors have $\epsilon$ precision, i.e., each entries is an integer multiple of $\epsilon$. The noise will be Gaussian with zero mean and variance $\sigma^2$, i.e., $\eta \sim \mathcal{N}(0, \sigma^2)$. Our main result of this section is the following.

**Theorem 3.** *It is possible to recover approximations with the $\ell_1/\ell_1$ guarantee in Eq. (3) with probability at least $1 - 2/n$ of all the unknown vectors $\beta^\ell \in \{0, \pm\epsilon, \pm2\epsilon, \pm3\epsilon, \ldots\}^n, \ell = 1, \ldots, L$ with $O(k(\log^3 n) \exp((\sigma/\epsilon)^{2/3})$ oracle queries where $\mathsf{SNR} = O(1/\sigma^2)$.*

Before we proceed with the ideas of proof, it would be useful to recall the *restricted isometry property* (RIP) of matrices in the context of recovery guarantees of (3). A matrix $\Phi \in \mathbb{R}^{m \times n}$ satisfies the $(k, \delta)$-RIP if for any vector $\mathbf{z} \in \mathbb{R}^n$ with $\|\mathbf{z}\|_0 \leq k$,

$$(1 - \delta)\|\mathbf{z}\|_2^2 \leq \|\Phi\mathbf{z}\|_2^2 \leq (1 + \delta)\|\mathbf{z}\|_2^2. \tag{5}$$

It is known that if a matrix is $(2k, \delta)$-RIP with $\delta < \sqrt{2} - 1$, then the guarantee of (3) (in particular, $\ell_1/\ell_1$-guarantee and also an $\ell_2/\ell_1$-guarantee) is possible [6] with the *the basis pursuit* algorithm, an efficient algorithm based on linear programming. It is also known that a random $\pm 1$ matrix (with normalized columns) satisfies the property with $c_s k \log n$ rows, where $c_s$ is an absolute constant [3].

There are several key ideas of the proof. Since the case of $L = 2$ is simpler to handle, we start with that and then provide the extra steps necessary for the general case subsequently.

**Algorithm 2** `Noisy Recovery for` $L = 2$ The algorithm for recovering best $k$-sparse approximation of vectors via queries to oracle in noisy setting.

---

**Require:** SNR $= 1/\sigma^2$, Precision of unknown vectors $\epsilon$, and the constant $c_s$ where $c_s k \log n$ rows are sufficient for RIP in binary matrices.
 1: **for** $i = 1, 2, \ldots, c_s k \log(n/k)$ **do**
 2:     Call SampleAndRecover($\mathbf{v}_i$) where $\mathbf{v}_i \in_R \{+1, -1\}^n$.
 3: **end for**
 4: **for** $i \in [\log n]$ and $j \in [c_s k \log(n/k)]$ with $j \neq i$ **do**
 5:     Call SampleAndRecover($(\mathbf{v}_i + \mathbf{v}_j)/2$) and SampleAndRecover($(\mathbf{v}_i - \mathbf{v}_j)/2$)
 6: **end for**
 7: Choose vector $\mathbf{v}$ from $\{\mathbf{v}_1, \mathbf{v}_2, \ldots, \mathbf{v}_{\log n}\}$ such that $\langle \mathbf{v}, \beta^1 \rangle \neq \langle \mathbf{v}, \beta^2 \rangle$.
 8: **for** $i = 1, 2, \ldots, k \log(n/k)$ and $\mathbf{v}_i \neq \mathbf{v}$ **do**
 9:     Label one of $\langle \mathbf{v}_i, \beta^1 \rangle, \langle \mathbf{v}_i, \beta^2 \rangle$ to be $\langle \mathbf{v}, \beta^1 \rangle$ if their sum is in the pair $\langle \frac{\mathbf{v}_i + \mathbf{v}}{2}, \beta^1 \rangle, \langle \frac{\mathbf{v}_i + \mathbf{v}}{2}, \beta^2 \rangle$ and their difference is in the pair $\langle \frac{\mathbf{v} - \mathbf{v}_i}{2}, \beta^1 \rangle, \langle \frac{\mathbf{v} - \mathbf{v}_i}{2}, \beta^2 \rangle$. Label the other $\langle v, \beta^2 \rangle$.
10: **end for**
11: Aggregate all (query, denoised query response pairs) labelled $\langle \mathbf{v}, \beta^1 \rangle$ and $\langle \mathbf{v}, \beta^2 \rangle$ separately and multiply all denoised query responses by a factor of $1/(\sqrt{c_s k \log(n/k)})$.
12: Return best $k$-sparse approximation of $\beta^1$ and $\beta^2$ by using Basis Pursuit algorithm on each aggregated cluster of (query, denoised query response) pairs.

13: **function** SampleAndRecover ($\boldsymbol{v}$)
14:     Issue $T = c_2 \exp\left((\sigma/\epsilon)^{2/3}\right)$ queries to oracle with $\boldsymbol{v}$.
15:     Return $\langle \boldsymbol{v}, \beta^1 \rangle$, $\langle \boldsymbol{v}, \beta^2 \rangle$ via min-distance estimator (Gaussian mixture learning, lemma 2).
16: **end function**

---

### 3.1 Gaussian Noise: Two vectors

Algorithm 2 addresses the setting with only two unknown vectors. We will assume $\|\beta^1\|_2 = \|\beta^2\|_2 = 1$, so that we can subsequently show that the SNR is simply $1/\sigma^2$. This assumption is not necessary but we make this for the ease of presentation. The assumption of $\epsilon$-precision for $\beta$ was made in Yin et al. [27], and we stick to the same assumption. On the other hand, Yin et al. requires further assumptions that we do not need to make. Furthermore, the result of Yin et al. is restricted to exactly sparse vectors, whereas our result holds for general sparse approximation.

For the two-vector case the result we aim to show is following.

**Theorem 4.** *Algorithm 2 uses* $O(k \log^3 n \exp((\sigma/\epsilon)^{2/3}))$ *queries to recover both the vectors* $\beta^1$ *and* $\beta^2$ *with an* $\ell_1/\ell_1$ *guarantee in Eq.* (3) *with probability at least* $1 - 2/n$.

This result is directly comparable with [27]. On the statistical side, we improve their result in several ways: (1) we improve the dependence on $\sigma/\epsilon$ in the sample complexity from $\exp(\sigma/\epsilon)$ to $\exp((\sigma/\epsilon)^{2/3})$,[3] (2) our result applies for dense vectors, recovering the best $k$-sparse approximations, and (3) we do not need the overlap assumption (eq. (2)) used in their work.

Once we show SNR $= 1/\sigma^2$, Theorem 4 trivially implies Theorem 3 in the case $L = 2$. Indeed, from Algorithm 2, notice that we have used vectors $\mathbf{v}$ sampled uniformly at random from $\{+1, -1\}^n$ and use them as query vectors. We must have $\mathbb{E}_\mathbf{v} |\langle \mathbf{v}, \beta^\ell \rangle|^2 / \mathbb{E}\eta^2 = \|\beta^\ell\|_2^2/\sigma^2 = 1/\sigma^2$ for $\ell = 1, 2$. Further, we have used the sum and difference query vectors which have the form $(\mathbf{v_1} + \mathbf{v_2})/2$ and $(\mathbf{v_1} - \mathbf{v_2})/2$ respectively where $v_1, v_2$ are sampled uniformly and independently from $\{+1, -1\}^n$. Therefore, we must have for $\ell = 1, 2$, $\mathbb{E}_{\mathbf{v_1}, \mathbf{v_2}} |\langle (\mathbf{v_1} \pm \mathbf{v_2})/2, \beta^\ell \rangle|^2 / \mathbb{E}\eta^2 = 1/2\sigma^2$. According to our definition of SNR, we have that SNR $= 1/\sigma^2$.

A description of Algorithm 2 that lead to proof of Theorem 4 can be found in Appendix B. We provide a short sketch here and state an important lemma that we will use in the more general case.

The main insight is that for a fixed sensing vector $\boldsymbol{v}$, if we repeatedly query with $\boldsymbol{v}$, we obtain samples from a mixture of Gaussians $\frac{1}{2}\mathcal{N}(\langle \boldsymbol{v}, \beta^1 \rangle, \sigma^2) + \frac{1}{2}\mathcal{N}(\langle \boldsymbol{v}, \beta^2 \rangle, \sigma^2)$. If we can *exactly* recover the

means of these Gaussians, we essentially reduce to the noiseless case from the previous section. The first key step upper bounds the sample complexity for exactly learning the parameters of a mixture of Gaussians.

**Lemma 2** (Learning Gaussian mixtures)**.** *Let $\mathcal{M} = \frac{1}{L}\sum_{i=1}^{L}\mathcal{N}(\mu_i, \sigma^2)$ be a uniform mixture of $L$ univariate Gaussians, with known shared variance $\sigma^2$ and with means $\mu_i \in \epsilon\mathbb{Z}$. Then, for some constant $c > 0$ and some $t = \omega(L)$, there exists an algorithm that requires $ctL^2 \exp((\sigma/\epsilon)^{2/3})$ samples from $\mathcal{M}$ and exactly identifies the parameters $\{\mu_i\}_{i=1}^{L}$ with probability at least $1 - 2e^{-2t}$.*

If we sense with $v \in \{-1, +1\}^n$ then $\langle v, \beta^1 \rangle, \langle v, \beta^2 \rangle \in \epsilon\mathbb{Z}$, so appealing to the above lemma, we can proceed assuming we know these two values exactly. Unfortunately, the sensing vectors here are more restricted — we must maintain bounded SNR and our technique of mixture learning requires that the means have finite precision — so we cannot simply appeal to our noiseless results for the alignment step. Instead we design a new alignment strategy, inspired by error correcting codes. Given two query vectors $v_1, v_2$ and the exact means $\langle v_i, \beta^j \rangle$, $i, j \in \{1, 2\}$, we must identify which values correspond to $\beta^1$ and $\beta^2$. In addition to sensing with any pair $v_1$ and $v_2$ we sense with $\frac{v_1 \pm v_2}{2}$, and we use these two additional measurements to identify which recovered means correspond to $\beta^1$ and which correspond to $\beta^2$. Intuitively, we can check if our alignment is correct via these reference measurements.

Therefore, we can obtain aligned, denoised inner products with each of the two parameter vectors. At this point we can apply a standard compressed sensing result as mentioned at the start of this section to obtain the sparse approximations of vectors.

## 3.2 General value of $L$

In this setting, we will have $L > 2$ unknown vectors $\beta^1, \beta^2, \ldots, \beta^L \in \mathbb{R}^n$ of unit norm each from which the oracle can sample from with equal probability. We assume that $L$ does not grow with $n$ or $k$ and as before, all the elements in the unknown vectors lie on a $\epsilon$-grid. Here, we will build on the ideas for the special case of $L = 2$.

The main result of this section is the following.

**Theorem 5.** *Algorithm 3 uses $O\left(k(\log n)^3 \exp\left(\left(\frac{\sigma}{\epsilon}\right)^{2/3}\right)\right)$ queries with $\mathsf{SNR} = O(1/\sigma^2)$ to recover all the vectors $\beta^1, \ldots, \beta^L$ with $\ell_1/\ell_1$ guarantees in Eq. (3) with probability at least $1 - 2/n$.*

Theorem 3 follows as a corollary of this result.

The analysis of Algorithm 3 and the proofs of Theorems 3 and 5 are provided in detail in Appendix D. Below we sketch some of the main points of the proof.

There are two main hurdles in extending the steps explained for $L = 2$. For a query vector $\mathbf{v}$, we define the *denoised query means* to be the set of elements $\{\langle \mathbf{v}, \beta^i \rangle\}_{i=1}^{L}$. Recall that a query vector $\mathbf{v}$ is defined to be *good* if all the elements in the set of denoised query means $\{\langle \mathbf{v}, \beta^1 \rangle, \langle \mathbf{v}, \beta^2 \rangle, \ldots, \langle \mathbf{v}, \beta^L \rangle\}$ are distinct. For $L = 2$, the probability of a query vector $\mathbf{v}$ being *good* for $L = 2$ is at least $1/2$ but for a value of $L$ larger than $2$, it is not possible to obtain such guarantees without further assumptions. For a more concrete example, consider $L \geq 4$ and the unknown vectors $\beta^1, \beta^2, \ldots, \beta^L$ to be such that $\beta^i$ has $1$ in the $i^{th}$ position and zero everywhere else. If $\mathbf{v}$ is sampled from $\{+1, -1\}^n$ as before, then $\langle \mathbf{v}, \beta^i \rangle$ can take values only in $\{-1, 0, +1\}$ and therefore it is not possible that all the values $\langle \mathbf{v}, \beta^i \rangle$ are distinct. Secondly, even if we have a *good* query vector, it is no longer trivial to extend the clustering or alignment step. Hence a number of new ideas are necessary to solve the problem for any general value of $L$.

We need to define a few constants which are used in the algorithm. Let $\delta < \sqrt{2} - 1$ be a constant (we need a $\delta$ that allow $k$-sparse approximation given a $(2k, \delta)$-RIP matrix). Let $c'$ be a large positive constant such that

$$\frac{\delta^2}{16} - \frac{\delta^3}{48} - \frac{1}{c'} > 0. \tag{A}$$

Secondly, let $\alpha^\star$ be another positive constant that satisfies the following for a given value of $c'$,

$$\alpha^\star = \max\left\{\alpha : \frac{\alpha^\alpha}{(\alpha - 1)^{\alpha - 1}} < \exp\left(\frac{\delta^2}{16} - \frac{\delta^3}{48} - \frac{1}{c'}\right)\right\}. \tag{B}$$

**Algorithm 3** `Noisy Recovery for any constant` $L$ The algorithm for recovering best $k$-sparse approximation of vectors via queries to oracle in noisy setting.

---

**Require:** $c', \alpha^\star, z^\star$ as defined in equations (A), (B) and (C) respectively, Variance of noise $\mathbb{E}\eta^2 = \sigma^2$ and precision of unknown vectors as $\epsilon$.

1: **for** $i = 1, 2, \ldots, \sqrt{\alpha^\star} \log n + c' \alpha^\star k \log(n/k)$ **do**
2:      Let $\mathbf{v}_i \in_R \{+1, -1\}^n$, $\mathbf{r}_i \in_R \{-2z^\star, -2z^\star + 1, \ldots, 2z^\star\}^n$, $q_i \in_R \{1, 2, \ldots, 4z^\star + 1\}$
3:      Make $c_2 \exp((\sigma/\epsilon)^{2/3})$ queries to the oracle using each of the vectors $(q_i - 1)\mathbf{r_i}, \mathbf{v_i} + q_i \mathbf{r_i}$ and $\mathbf{v_i} + \mathbf{r_i}$.
4:      Recover $\langle \{(q_i - 1)\mathbf{r}_i, \beta^t\} \rangle_{t=1}^L, \{\langle \mathbf{v_i} + \mathbf{r_i}, \beta^t \rangle\}_{t=1}^L, \{\langle \mathbf{v}_i + q_i \mathbf{r}_i, \beta^t \rangle\}_{t=1}^L$ by using min-distance estimator (Gaussian mixture learning, lemma 2).
5: **end for**
6: **for** $i \in [\sqrt{\alpha^\star} \log n]$ and $j \in [\alpha^\star k \log(nk)]$ **do**
7:      Make $c_2 \exp((\sigma/\epsilon)^{2/3})$ queries to the oracle using the vector $\mathbf{r}_{i+j} + \mathbf{r}_i$.
8:      Recover $\{\langle \mathbf{r}_{i+j} + \mathbf{r}_i, \beta^t \rangle\}_{t=1}^L$, by using the min-distance estimator (Gaussian mixture learning, Lemma 2).
9: **end for**
10: Choose vector $(\mathbf{v}^\star, \mathbf{r}^\star, q^\star)$ from $\{(\mathbf{v}_t, \mathbf{r}_t, q_t)\}_{t=1}^{\sqrt{\alpha^\star} \log n}$ such that $(\mathbf{v}^\star + \mathbf{r}^\star, (q-1)\mathbf{r}^\star, \mathbf{v}^\star + q^\star \mathbf{r}^\star)$ is good. Call a triplet $(\mathbf{v} + \mathbf{r}, (q - 1)\mathbf{r}, \mathbf{v} + q\mathbf{r})$ to be good if no element in $\{\langle \mathbf{v} + q\mathbf{r}, \beta^i \rangle\}_{i=1}^L$ can be written in two possible ways as sum of two elements, one each from $\{\langle \mathbf{v} + \mathbf{r}, \beta^i \rangle\}_{i=1}^L$ and $\{\langle (q-1)\mathbf{r}, \beta^i \rangle\}_{i=1}^L$.
11: Initialize $\mathcal{S}_j = \phi$ for $j = 1, \ldots, L$
12: **for** $i = \sqrt{\alpha^\star} \log n + 1, 2, \ldots, \sqrt{\alpha^\star} \log n + c' \alpha^\star k \log \frac{n}{k}$ **do**
13:      **if** $(\mathbf{v}_i + \mathbf{r}_i, (q_i - 1)\mathbf{r}_i, \mathbf{v}_i + q\mathbf{r}_i)$ is matching good with respect to $(\mathbf{v}^\star + \mathbf{r}^\star, (q-1)\mathbf{r}^\star, \mathbf{v}^\star + q^\star \mathbf{r}^\star)$ (Call a triplet $(\mathbf{v}' + \mathbf{r}', (q' - 1)\mathbf{r}', \mathbf{v}' + q'\mathbf{r}')$ to be matching good w.r.t a good triplet $(\mathbf{v}^\star + \mathbf{r}^\star, (q^\star - 1)\mathbf{r}^\star, \mathbf{v}^\star + q^\star \mathbf{r}^\star)$ if $(\mathbf{v}' + \mathbf{r}', (q' - 1)\mathbf{r}', \mathbf{v}' + q'\mathbf{r}')$ and $(\mathbf{r}', \mathbf{r}^\star, \mathbf{r}' + \mathbf{r}^\star)$ are good. ) **then**
14:          Label the elements in $\{\langle \mathbf{v}_i, \beta^t \rangle\}_{t=1}^L$ as described in Lemma 18
15:          **for** $j = 1, 2, \ldots, L$ **do**
16:             $\mathcal{S}_j = \mathcal{S}_j \cup \{\langle \mathbf{v}_i, \beta^t \rangle\}$ if label of $\langle \mathbf{v}_i, \beta^t \rangle$ is $\langle \mathbf{r}^\star, \beta^j \rangle$
17:          **end for**
18:      **end if**
19: **end for**
20: **for** $j = 1, 2, \ldots, L$ **do**
21:      Aggregate the elements of $\mathcal{S}_j$ and scale them by a factor of $1/c'k \log(n/k)$.
22:      Recover the vector $\beta^j$ by using basis pursuit algorithms (compressed sensing decoding).
23: **end for**
24: Return $\beta^1, \beta^2, \ldots, \beta^L$.

---

Finally, for a given value of $\alpha^\star$ and $L$, let $z^\star$ be the smallest integer that satisfies the following:

$$z^\star = \min \left\{ z \in \mathbb{Z} : 1 - L^3 \left( \frac{3}{4z + 1} - \frac{1}{4z^2 + 1} \right) \geq \frac{1}{\sqrt{\alpha^\star}} \right\}. \tag{C}$$

**The Denoising Step.** In each step of the algorithm, we sample a vector $\mathbf{v}$ uniformly at random from $\{+1, -1\}^n$, another vector $\mathbf{r}$ uniformly at random from $\mathcal{G} \equiv \{-2z^\star, -2z^\star + 1, \ldots, 2z^\star - 1, 2z^\star\}^n$ and a number $q$ uniformly at random from $\{1, 2, \ldots, 4z^\star + 1\}$. Now, we will use a batch of queries corresponding to the vectors $\mathbf{v} + \mathbf{r}, (q - 1)\mathbf{r}$ and $\mathbf{v} + q\mathbf{r}$. We define a triplet of query vectors $(\mathbf{v}_1, \mathbf{v}_2, \mathbf{v}_3)$ to be *good* if for all triplets of indices $i, j, k \in [L]$ such that $i, j, k$ are not identical,

$$\langle \mathbf{v}_1, \beta^i \rangle + \langle \mathbf{v}_2, \beta^j \rangle \neq \langle \mathbf{v}_3, \beta^k \rangle.$$

We show that the query vector triplet $(\mathbf{v} + \mathbf{r}, (q - 1)\mathbf{r}, \mathbf{v} + q\mathbf{r})$ is good with at least some probability. This implies if we choose $O(\log n)$ triplets of such query vectors, then at least one of the triplets are good with probability $1 - 1/n$. It turns out that, for a good triplet of vectors $(\mathbf{v} + \mathbf{r}, (q - 1)\mathbf{r}, \mathbf{v} + q\mathbf{r})$, we can obtain $\langle \mathbf{v}, \beta^i \rangle$ for all $i \in [L]$.

Furthermore, it follows from Lemma 2 that for a query vector $\mathbf{v}$ with integral entries, a batch size of $T > c_3 \log n \exp((\sigma/\epsilon)^{2/3})$, for some constant $c_3 > 0$, is sufficient to recover the denoised query responses $\langle \mathbf{v}, \beta^1 \rangle, \langle \mathbf{v}, \beta^2 \rangle, \ldots, \langle \mathbf{v}, \beta^L \rangle$ for all the queries with probability at least $1 - 1/\text{poly}(n)$.

**The Alignment Step.** Let a particular good query vector triplet be $(\mathbf{v}^\star + \mathbf{r}^\star, (q^\star - 1)\mathbf{r}^\star, \mathbf{v}^\star + q^\star \mathbf{r}^\star)$. From now, we will consider the $L$ elements $\{\langle \mathbf{r}^\star, \beta^i \rangle\}_{i=1}^L$ to be labels and for a vector $\mathbf{u}$, we will associate a label with every element in $\{\langle \mathbf{u}, \beta^i \rangle\}_{i=1}^L$. The labelling is correct if, for all $i \in [L]$, the element labelled as $\langle \mathbf{r}^\star, \beta^i \rangle$ also corresponds to the same unknown vector $\beta^i$. Notice that we can label the elements $\{\langle \mathbf{v}^\star, \beta^i \rangle\}_{i=1}^L$ correctly because the triplet $(\mathbf{v}^\star + \mathbf{r}^\star, (q^\star - 1)r^\star, \mathbf{v}^\star + q^\star \mathbf{r}^\star)$ is good. Consider another good query vector triplet $(\mathbf{v}' + \mathbf{r}', (q' - 1)\mathbf{r}', \mathbf{v}' + q'\mathbf{r}')$. This *matches* with the earlier query triplet if additionally, the vector triplet $(\mathbf{r}', \mathbf{r}^\star, \mathbf{r}' + \mathbf{r}^\star)$ is also good.

Such matching pair of good triplets exists, and can be found by random choice with some probability. We show that, the matching good triplets allow us to do the alignment in the case of general $L > 2$.

At this point we would again like to appeal to the standard compressed sensing results. However we need to show that the matching good vectors themselves form a matrix that has the required RIP property. As our final step, we establish this fact.

**Remark 3** (Refinement and adaptive queries)**.** *It is possible to have a sample complexity of* $O\left(k(\log n)^2 \log k \exp\left((\epsilon\sqrt{\mathsf{SNR}})^{-2/3}\right)\right)$ *in Theorem 3, but with a probability of* $1 - \mathrm{poly}(k^{-1})$. *Also it is possible to shave-off another* $\log n$ *factor from sample complexity if we can make the queries adaptive.*

*Acknowledgements:* This research is supported in part by NSF Grants CCF 1642658, 1618512, 1909046, 1908849 and 1934846.

## Footnotes

[1]Many of our results can be generalized to non-uniform distributions but we will assume a uniform distribution throughout for the sake of clarity.

[2] Note that we do not assume $\epsilon$-precision in the noiseless case.

[3]Note that [27] treat $\sigma/\epsilon$ as constant in their theorem statement, but the dependence can be extracted from their proof.

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
