[Supplementary Material]

# Supplementary Material: Sample Complexity of Learning Mixtures of Sparse Linear Regressions

## A  Proof of Theorem 2

It is known that for any particular vector $\beta$, at least $2k$ queries to the oracle are necessary in order to recover the vector exactly. Suppose the random variable $X$ denotes the number of queries until the oracle has sampled the vector $\beta$ at least $2k$ times. Notice that $X = \sum_{i=1}^{2k} X_i$ can be written as a sum of independent and identical random variables $X_i$ distributed according to the geometric distribution with parameter $1/L$ where $X_i$ denotes the number of attempts required to obtain the $i^{\text{th}}$ sample after the $(i-1)^{\text{th}}$ sample has been made by the oracle. Since $X$ is a sum of independent random variables, we must have

$$\mathbb{E}X = 2Lk \quad \text{and} \quad \mathrm{Var}(X) = 2k(L^2 - L)$$

Therefore by using Chebychev's inequality [5], we must have

$$\Pr\left(X \le 2Lk - k^{\frac{1}{4}}\sqrt{2k(L^2 - L)}\right) \le \frac{1}{\sqrt{k}}$$

and therefore $X > 2Lk(1 - o(1))$ with high probability which proves the statement of the theorem.

## B  Description of Algorithm 2 and Proof of Theorem 4

**Algorithm 2 (Design of queries and denoising):** Let $m$ be the total number of queries that we will make. In the first step of the algorithm, for a particular query vector $\mathbf{v} \in \mathbb{R}^n$, our objective is to recover $\langle \mathbf{v}, \beta^1 \rangle$ and $\langle \mathbf{v}, \beta^2 \rangle$ which we will denote as the *denoised query responses* corresponding to the vector $\mathbf{v}$. It is intuitive, that in order to do this, we need to use the same query vector $\mathbf{v}$ repeatedly a number of times and aggregate the noisy query responses to recover the denoised counterparts.

Therefore, at every iteration in Step 1 of Algorithm 2, we sample a vector $\mathbf{v}$ uniformly at random from $\{+1, -1\}^n$. Once the vector $\mathbf{v}$ is sampled, we use $\mathbf{v}$ as query vector repeatedly for $T$ times. We will say that the query responses to the same vector as query to be a *batch* of size $T$. It can be seen that since $\mathbf{v}$ is fixed, the query responses in a batch is sampled from a Gaussian mixture distribution $\mathcal{M}$ with means $\langle \mathbf{v}, \beta^1 \rangle$ and $\langle \mathbf{v}, \beta^2 \rangle$ and variance $\sigma^2$, in short,

$$\mathcal{M} = \frac{1}{2}\mathcal{N}(\langle \mathbf{v}, \beta^1 \rangle, \sigma^2) + \frac{1}{2}\mathcal{N}(\langle \mathbf{v}, \beta^2 \rangle, \sigma^2).$$

Therefore the problem reduces to recovering the mean parameters from a mixture of Gaussian distribution with at most two mixture constituents (since the means can be same) and having the same variance. We will use the following important lemma for this problem.

**Lemma** (Lemma 2: Learning Gaussian mixtures)**.** *Let $\mathcal{M} = \frac{1}{L}\sum_{i=1}^{L}\mathcal{N}(\mu_i, \sigma^2)$ be a uniform mixture of $L$ univariate Gaussians, with known shared variance $\sigma^2$ and with means $\mu_i \in \epsilon\mathbb{Z}$. Then, for some constant $c > 0$ and some $t = \omega(L)$, there exists an algorithm that requires $ctL^2 \exp((\sigma/\epsilon)^{2/3})$ samples from $\mathcal{M}$ and exactly identifies the parameters $\{\mu_i\}_{i=1}^{L}$ with probability at least $1 - 2e^{-2t}$.*

The proof of this lemma can be found in Appendix C. We now have the following lemma to characterize the size of each batch $T$.

**Lemma 4.** *For any query vector $\mathbf{v} \in \{+1, 0, -1\}^n$, a batchsize of $T = c_1 \log n \exp((\sigma/\epsilon)^{2/3})$, for a constant $c_1 > 0$, is sufficient to recover the denoised query responses $\langle \mathbf{v}, \beta^1 \rangle$ and $\langle \mathbf{v}, \beta^2 \rangle$ with probability at least $1 - 1/\mathrm{poly}(n)$.*

*Proof.* Since $\mathbf{v} \in \{+1, 0, -1\}^n$, $\langle \mathbf{v}, \beta^1 \rangle, \langle \mathbf{v}, \beta^2 \rangle \in \epsilon\mathbb{Z}$. Using Lemma 2, the claim follows. $\qquad\square$

**Corollary 5.** *For any $O\left(k \log n \log(n/k)\right)$ query vectors sampled uniformly at random from $\{+1, -1\}^n$, a batch size of $T > c_2 \log n \exp((\frac{\sigma}{\epsilon})^{2/3})$, for some constant $c_2 > 0$, is sufficient to recover the denoised query responses corresponding to every query vector with probability at least $1 - 1/\mathrm{poly}(n)$.*

*Proof.* This statement is proved by taking a union bound over $O\left(k \log n \log(n/k)\right)$ batches corresponding to that many query vectors. $\qquad\square$

**Algorithm 2 (Alignment step):** Notice from the previous discussion, for each batch corresponding to a query vector $\mathbf{v}$, we obtain the pair of values $(\langle \mathbf{v}, \beta^1 \rangle, \langle \mathbf{v}, \beta^2 \rangle)$. However, we still need to cluster these values (by taking one value from each pair and assigning it to one of the clusters) into two clusters corresponding to $\beta_1$ and $\beta_2$. We will first explain the clustering process for two particular query vectors $\mathbf{v}_1$ and $\mathbf{v}_2$ for which we have already obtained the pairs $(\langle \mathbf{v}_1, \beta^1 \rangle, \langle \mathbf{v}_1, \beta^2 \rangle)$ and $(\langle \mathbf{v}_2, \beta^1 \rangle, \langle \mathbf{v}_2, \beta^2 \rangle)$. The objective is to cluster the four samples into two groups of two samples each so that the samples in each cluster correspond to the same unknown sensed vector. Now, we have two cases to consider:

**Case 1:** $(\langle \mathbf{v}_1, \beta^1 \rangle = \langle \mathbf{v}_1, \beta^2 \rangle$ or $\langle \mathbf{v}_2, \beta^1 \rangle = \langle \mathbf{v}_2, \beta^2 \rangle)$ In this scenario, the values in at least one of the pairs are same and any grouping works.

**Case 2:** $(\langle \mathbf{v}_1, \beta^1 \rangle \neq \langle \mathbf{v}_1 \beta^2 \rangle$ and $\langle \mathbf{v}_2, \beta^1 \rangle \neq \langle \mathbf{v}_2, \beta^2 \rangle)$. We use two more batches corresponding to the vectors $\frac{\mathbf{v}_1 + \mathbf{v}_2}{2}$ and $\frac{\mathbf{v}_1 - \mathbf{v}_2}{2}$ which belong to $\{-1, 0, +1\}^n$. We will call the vector $\frac{\mathbf{v}_1 + \mathbf{v}_2}{2}$ the *sum query* and the vector $\frac{\mathbf{v}_1 - \mathbf{v}_2}{2}$ the *difference query* corresponding to $\mathbf{v}_1, \mathbf{v}_2$ respectively. Hence using Lemma 4 again, we will be able to obtain the pairs $(\langle \frac{\mathbf{v}_1 + \mathbf{v}_2}{2}, \beta^1 \rangle, \langle \frac{\mathbf{v}_1 + \mathbf{v}_2}{2}, \beta^2 \rangle)$ and $(\langle \frac{\mathbf{v}_1 - \mathbf{v}_2}{2}, \beta^1 \rangle, \langle \frac{\mathbf{v}_1 - \mathbf{v}_2}{2}, \beta^2 \rangle)$. Now, we will choose two elements from the pairs $(\langle \mathbf{v}_1, \beta^1 \rangle, \langle \mathbf{v}_1 \beta^2 \rangle)$ and $(\langle \mathbf{v}_2, \beta^1 \rangle, \langle \mathbf{v}_2 \beta^2 \rangle)$ (one element from each pair) such that their sum belongs to the pair $2\langle \frac{\mathbf{v}_1 + \mathbf{v}_2}{2}, \beta^1 \rangle, 2\langle \frac{\mathbf{v}_1 + \mathbf{v}_2}{2}, \beta^2 \rangle$ and their difference belongs to the pair $2\langle \frac{\mathbf{v}_1 - \mathbf{v}_2}{2}, \beta^1 \rangle, 2\langle \frac{\mathbf{v}_1 - \mathbf{v}_2}{2}, \beta^2 \rangle$. In our algorithm, we will put these two elements into one cluster and the other two elements into the other cluster. From construction, we must put $(\langle \mathbf{v}_1, \beta^1 \rangle, \langle \mathbf{v}_2, \beta^1 \rangle)$ in one cluster and $(\langle \mathbf{v}_1, \beta^2 \rangle, \langle \mathbf{v}_2, \beta^2 \rangle)$ in other.

Putting it all together, in Algorithm 2, we uniformly and randomly choose $c_s k \log \frac{n}{k}$ query vectors from $\{+1, -1\}^n$ and for each of them, we use it repeatedly for $c_2 \log n \exp\left(\frac{\sigma}{\epsilon}\right)^{2/3}$ times. From each batch, we recover the denoised query responses for the query vector associated with that batch. For a particular query vector $\mathbf{v}$, we call the query vector *good* if $\langle \mathbf{v}, \beta^1 \rangle \neq \langle \mathbf{v}, \beta^2 \rangle$. For a $\mathbf{v}$ chosen uniformly at randomly from $\{+1, -1\}^n$, the probability that $\langle \mathbf{v}, \beta^1 - \beta^2 \rangle = 0$ is at most $\frac{1}{2}$. Therefore, if one chooses $\log n$ query vectors uniformly and independently at random from $\{+1, -1\}^n$, at least one is good with probability $1 - \frac{1}{n}$. We are now ready to prove the main theorem.

*Proof of Theorem 4.* For each vector $\mathbf{v}$ belonging to the set of first $\log n$ query vectors and for each query vector $\mathbf{b}$ ($\mathbf{b}$ is among the initial $c_s k \log \frac{n}{k}$ query vectors) different from $\mathbf{v}$, we make two additional batches of queries corresponding to query vectors $\frac{\mathbf{v} + \mathbf{b}}{2}$ and $\frac{\mathbf{v} - \mathbf{b}}{2}$. Consider the first $\log n$ query vectors. We know that one of them, say $\mathbf{g}$, is a good query vector. Let us denote the denoised means obtained from the batch of queries corresponding to $\mathbf{g}$ to be $(x, y)$. We can think of $x$ and $y$ as labels for the clustering of the denoised means from the other query vectors. Now, from the alignment step, we know that for every query vector $\mathbf{b}$ different from $\mathbf{g}$ and the denoised query responses $(p, q)$ corresponding to $\mathbf{b}$, by using the additional sum and difference queries, we can label one of the element in $(p, q)$ as $x$ and the other one as $y$. Since the vector $\mathbf{g}$ is good, therefore $x \neq y$ and hence we will be able to aggregate the denoised query responses corresponding to $\beta^1$ and the denoised query responses corresponding to $\beta^2$ separately. Since we have $c_s k \log n$ query responses for each of $\beta^1$ and $\beta^2$, we can scale the query responses by a factor of $1/\sqrt{c_s k \log n}$ and subsequently, we can run basis pursuit [6] to recover the best $k$-sparse approximations of both $\beta^1$ and $\beta^2$. Notice that the total number of queries in this scheme is $O(k \log^2 n)$ and since the size of each batch corresponding to each query is $O(\log n \exp((\frac{\sigma}{\epsilon})^{2/3}))$, the total sample complexity required is $O\left(k(\log n)^3 \exp\left(\frac{\sigma}{\epsilon}\right)^{2/3}\right)$. $\qquad\square$

## C  Proof of Lemma 2

Note that Lemma 2 is *not* a claimed contribution of this paper. Rather, it appears as one of the results in another submission (to a different conference). Since we can't cite this other paper yet we include the details here for completeness.

**Lemma 6.** *For any two distributions $f, f'$ defined over the same sample space $\Omega \subseteq \mathbb{R}$, we have*

$$\|f - f'\|_{TV} \geq \frac{1}{2} \sup_{t \in \mathbb{R}} |C_f(t) - C'_f(t)|.$$

*More generally, for any $G : \Omega \to \mathbb{C}$ and $\Omega' \subset \Omega$ we have*

$$\|f - f'\|_{TV} \geq \left(2 \sup_{x \in \Omega'} |G(x)|\right)^{-1} \left(|\mathbb{E}_{X \sim f} G(X) - \mathbb{E}_{X \sim f'} G(X')|\right.$$
$$\left. - \int_{x \in \Omega \backslash \Omega'} |G(x)| \cdot |df(x) - df'(x)|\right).$$

*Proof.* We prove the latter statement, which implies the former since for the function $G(x) = e^{itx}$ we have $\sup_x |G(x)| = 1$. By the triangle inequality we have

$$|\mathbb{E}_{X \sim f} G(X) - \mathbb{E}_{X \sim f'} G(X)| \leq \int_{x \in \Omega} |G(x)| \cdot |df(x) - df'(x)|$$

$$\leq 2 \sup_{x \in \Omega'} |G(x)| \cdot \|f - f'\|_{TV} + \int_{x \in \Omega \backslash \Omega'} |G(x)| \cdot |df(x) - df'(x)|. \qquad \square$$

**Lemma 7.** *Let $z = \exp(it)$ where $t \in [-\pi/L, \pi/L]$. If the random variable $X \sim \mathcal{N}(\mu, \sigma)$ and $G_t(x) = e^{itx}$ then*

$$\mathbb{E}[G_t(X)] = \exp(-\sigma^2 t^2/2) z^\mu \text{ and } \|G_t\|_\infty = 1 .$$

*Proof.* Observe that $\mathbb{E}[G_t(X)]$ is precisely the characteristic function. Clearly we have $\|G_t\|_\infty = 1$ and further

$$\mathbb{E}[G_t(X)] = \exp(it\mu - \sigma^2 t^2/2) = \exp(-\sigma^2 t^2/2) z^\mu.$$

$$\square$$

We crucially use the following lemma.

**Lemma 8** ([4]). *Let $a_0, a_1, a_2, \dots \in \{-L, -(L-1), \dots, L-1, L\}$ be such that not all of them are zero. For any complex number $z$, let $A(z) \equiv \sum_\ell a_\ell z^\ell$. Then, for some absolute constant $c$,*

$$\max_{-\pi/S \leq t \leq \pi/S} |A(e^{it})| \geq e^{-cS} .$$

**Lemma 9** (TV Lower Bounds). *Consider two mixtures of Gaussian distributions such that $\mathcal{M} = \frac{1}{L} \sum_{i=1}^L \mathcal{N}(\mu_i, \sigma)$ and $\mathcal{M}' = \frac{1}{L} \sum_{i=1}^L \mathcal{N}(\mu'_i, \sigma)$ where $\mu_i, \mu'_i \in \epsilon\mathbb{Z}$. Then*

$$\|\mathcal{M}' - \mathcal{M}\|_{TV} \geq L^{-1} \exp(-\Omega((\sigma/\epsilon)^{2/3})).$$

*Proof.* The characteristic function of a Gaussian $X \sim \mathcal{N}(\mu, \sigma^2)$ is

$$C_\mathcal{N}(t) = \mathbb{E}e^{itX} = e^{it\mu - \frac{t^2 \sigma^2}{2}}.$$

Therefore we have that

$$C_\mathcal{M}(t) - C_{\mathcal{M}'}(t) \geq \frac{e^{-\frac{t^2 \sigma^2}{2}}}{L} \sum_{i=1}^L (e^{it\mu_i} - e^{it\mu'_i}).$$

Now, using Lemma 8, there exist an absolute constant $c$ such that,

$$\max_{-\frac{\pi}{\epsilon S} \leq t \leq \frac{\pi}{\epsilon S}} \left|\sum_{i=1}^L (e^{it\mu_i} - e^{it\mu'_i})\right| \geq e^{-cS}.$$

Also, for $t \in (-\frac{\pi}{\epsilon S}, \frac{\pi}{\epsilon S})$, $e^{-\frac{t^2 \sigma^2}{2}} \geq e^{-\frac{\sigma^2 \pi^2}{2\epsilon^2 S^2}}$. And therefore,

$$\left|C_\mathcal{M}(t) - C_{\mathcal{M}'}(t)\right| \geq \frac{1}{L} e^{-\frac{\sigma^2 \pi^2}{2\epsilon^2 S^2} - cS}.$$

By substituting $S = \frac{(\pi\sigma)^{2/3}}{(\epsilon^2 c)^{1/3}}$ above we conclude that there exists $t$ such that

$$\left| C_{\mathcal{M}}(t) - C_{\mathcal{M}'}(t) \right| \geq \frac{1}{L} e^{-\frac{3}{2}(c\pi\sigma/\epsilon)^{2/3}}.$$

Now using Lemma 6, we have $\|\mathcal{M}' - \mathcal{M}\|_{\mathrm{TV}} \geq L^{-1} \exp(-\Omega((\sigma/\epsilon)^{2/3}))$. $\qquad\square$

To learn the parameters of a Gaussian mixture

$$\mathcal{M} = \frac{1}{L} \sum_{i=1}^{L} \mathcal{N}(\mu_i, \sigma) \quad \text{where} \quad \mu_i \in \{\ldots, -2\epsilon, -\epsilon, 0, \epsilon, 2\epsilon \ldots\}$$

we use the minimum distance estimator precisely defined in [12, Section 6.8]. Let $\mathcal{A} \equiv \{\{x : \mathcal{M}(x) \geq \mathcal{M}'(x)\} : \text{for any two mixtures } \mathcal{M} \neq \mathcal{M}'\}$ be a collection of subsets. Let $P_m$ denote the empirical probability measure induced by the $m$ samples. Then, choose a mixture $\hat{\mathcal{M}}$ for which the quantity $\sup_{A \in \mathcal{A}} |\Pr_{\sim \hat{\mathcal{M}}}(A) - P_m(A)|$ is minimum (or within $1/m$ of the infimum). This is the minimum distance estimator, whose performance is guaranteed by the following proposition [12, Thm. 6.4].

**Lemma 10.** *Given $m$ samples from $\mathcal{M}$ and with $\Delta = \sup_{A \in \mathcal{A}} |\Pr_{\sim \mathcal{M}}(A) - P_m(A)|$, we have*

$$\left\| \hat{\mathcal{M}} - \mathcal{M} \right\|_{TV} \leq 4\Delta + \frac{3}{m}.$$

We now upper bound the right-hand side of the above inequality. It is known that the mean of $\Delta$ is bounded from above by a function of $VC(\mathcal{A})$, the VC dimension of the class $\mathcal{A}$, see [12, Section 4.3] and is given by

$$\mathbb{E}\Delta \leq c_2 \sqrt{\frac{VC(\mathcal{A})}{m}} \quad \text{for some universal constant } c_2 > 0$$

Now, via McDiarmid's inequality and a standard symmetrization argument, $\Delta$ is concentrated around its mean, see [12, Section 2.4]: and therefore, for some $t > 0$

$$\Delta \leq \mathbb{E}\Delta + \sqrt{\frac{t}{m}}$$

with probability at least $1 - 2e^{-2t}$. Therefore, we must have

$$\left\| \hat{\mathcal{M}} - \mathcal{M} \right\|_{TV} \leq 4\Delta + O(1/m) \leq 4\mathbb{E}_{\sim \mathcal{M}}\Delta + \sqrt{\frac{t}{m}} + o(1/\sqrt{m}) \leq 4\sqrt{\frac{VC(\mathcal{A})}{m}} + \sqrt{\frac{t}{m}},$$

with probability at least $1 - 2e^{-2t}$. This first term is bounded by the following:

**Lemma 11.** *For the class $\mathcal{A}$ defined above, the VC dimension is given by $VC(\mathcal{A}) = O(L)$.*

*Proof.* First of all we show that any element of the set $\mathcal{A}$ can be written as union of at most $4L - 1$ intervals in $\mathbb{R}$. For this we use the fact that a linear combination of $L$ Gaussian pdfs $f(x) = \sum_{i=1}^{L} \alpha_i f_i(x)$ where $f_i$s normal pdf $\mathcal{N}(\mu_i, \sigma_i^2)$ and $\alpha_i \in \mathbb{R}, 1 \leq i \leq L$ has at most $2L - 2$ zero-crossings [17]. Therefore, for any two mixtures of interest $\mathcal{M}(x) - \mathcal{M}'(x)$ has at most $4L - 2$ zero-crossings. Therefore any $A \in \mathcal{A}$ must be a union of at most $4L - 1$ contiguous regions in $\mathbb{R}$. It is now an easy exercise to see that the VC dimension of such a class is $\Theta(L)$. $\qquad\square$

As a result, when $t = \omega(L)$ the error of the minimum distance estimator is less $2\sqrt{\frac{t}{m}}$ with probability at least $1 - 2e^{-2t}$. But from lemma 9, notice that for any other mixture $\mathcal{M}'$ we must have,

$$\|\mathcal{M} - \mathcal{M}'\|_{\mathrm{TV}} \geq L^{-1} \exp(-\Omega((\sigma/\epsilon)^{2/3})).$$

As long as $\left\| \hat{\mathcal{M}} - \mathcal{M} \right\|_{\mathrm{TV}} \leq \frac{1}{2} \|\mathcal{M} - \mathcal{M}'\|_{\mathrm{TV}}$ we will exactly identify the parameters. Therefore, for some universal constant $c' > 0$, $m = c't L^2 \exp((\sigma/\epsilon)^{2/3})$ samples suffice to exactly learn the parameters with probability at least $1 - 2e^{-2t}$.

## D   Analysis of Algorithm 3 for General $L$ and Proof of Theorem 3 and Theorem 5

**Algorithm 3 (Design of queries):** In every iteration in Step 1 of Algorithm 3, we will sample a vector $\mathbf{v}$ uniformly at random from $\{+1, -1\}^n$, another vector $\mathbf{r}$ uniformly at random from $\mathcal{G} \equiv \{-2z^\star, -2z^\star + 1, \ldots, 2z^\star - 1, 2z^\star\}^n$ and a number $q$ uniformly at random from $\{1, 2, \ldots, 4z^\star + 1\}$. Now, we will use a batch of queries corresponding to the vectors $\mathbf{v} + \mathbf{r}, (q-1)\mathbf{r}$ and $\mathbf{v} + q\mathbf{r}$. We have the following lemmas describing several necessary properties of such queries.

We will define a triplet of query vectors $(\mathbf{v}_1, \mathbf{v}_2, \mathbf{v}_3)$ to be *good* if for all triplets of indices $i, j, k \in [L]$ such that $i, j, k$ are not identical, it must happen that

$$\langle \mathbf{v}_1, \beta^i \rangle + \langle \mathbf{v}_2, \beta^j \rangle \neq \langle \mathbf{v}_3, \beta^k \rangle$$

**Lemma 12.** *The query vector triplet $(\mathbf{v} + \mathbf{r}, (q-1)\mathbf{r}, \mathbf{v} + q\mathbf{r})$ is good with probability at least $\frac{1}{\sqrt{\alpha^\star}}$.*

*Proof.* Notice that for a fixed triplet $i, j, k \in [L]$ such that $i, j, k$ are not identical, we must have

$$
\begin{aligned}
&\Pr(\langle \mathbf{v} + \mathbf{r}, \beta^i \rangle + \langle (q-1)\mathbf{r}, \beta^j \rangle = \langle \mathbf{v} + q\mathbf{r}, \beta^k \rangle) \\
&= \Pr(\langle \mathbf{r}, \beta^i + (q-1)\beta^j - q\beta^k \rangle = \langle \mathbf{v}, \beta^k - \beta^i \rangle) \\
&\leq \Pr(\beta^i + (q-1)\beta^j - q\beta^k = 0) + \Pr(\beta^i + (q-1)\beta^j - q\beta^k \neq 0) \\
&\quad \cdot \Pr(\langle \mathbf{r}, \beta^i + (q-1)\beta^j - q\beta^k \rangle = \langle \mathbf{v}, \beta^k - \beta^i \rangle \mid \beta^i + (q-1)\beta^j - q\beta^k \neq 0) \\
&\leq \left(1 - \frac{1}{4z^\star + 1}\right) \frac{1}{4z^\star + 1} + \frac{1}{4z^\star + 1} = \frac{2}{4z^\star + 1} - \frac{1}{(4z^\star + 1)^2}.
\end{aligned}
$$

Notice that $\beta^i + (q-1)\beta^j - q\beta^k = 0$ cannot hold for two values of $q : q_1$ and $q_2$. We will show this fact by contradiction. Suppose it happens that $\beta^i + (q_1 - 1)\beta^j - q_1\beta^k = 0$ and $\beta^i + (q_2 - 1)\beta^j - q_2\beta^k = 0$ in which case we must have $\beta^j = \beta^k$ which is a contradiction to the fact that all the unknown vectors are distinct. We can take a union over all possible triplets (at most $L^3$ of them) and therefore we must have that

$$\Pr(\text{The vector triplet } (\mathbf{v} + \mathbf{r}, (q-1)\mathbf{r}, \mathbf{v} + q\mathbf{r}) \text{ is good}) \geq 1 - L^3 \left(\frac{2}{4z^\star + 1} - \frac{1}{(4z^\star + 1)^2}\right)$$

$$\geq \frac{1}{\sqrt{\alpha^\star}}.$$

$\square$

We will now generalize Lemma 4 in order to characterize the batch size required to recover the denoised query responses when there are $L$ unknown vectors that the oracle can sample from.

**Lemma 13** (Generalization of Lemma 4). *For a particular query vector $\mathbf{v}$ such that each entry of $\mathbf{v}$ is integral, a batch size of $T > c_3 \log n \exp((\sigma/\epsilon)^{2/3})$, for some constant $c_3 > 0$, is sufficient to recover the denoised query responses $\langle \mathbf{v}, \beta^1 \rangle, \langle \mathbf{v}, \beta^2 \rangle, \ldots, \langle \mathbf{v}, \beta^L \rangle$ with probability at least $1 - 1/\mathrm{poly}(n)$.*

*Proof.* The proof follows in exactly the same manner as the proof in Lemma 4 but in this case, we invoke Lemma 2 with any general value of $L$. Since we have assumed that $L$ is a constant, the term $L^2$ is subsumed within the constant $c_3$. $\square$

**Corollary 14.** *For $O\left(k \log^2 n\right)$ query vectors such that every entry of every query vector is integral, a batch size of $T > c_4 \log n \exp((\sigma/\epsilon)^{2/3})$, for some constant $c_4 > 0$, is sufficient to recover the denoised query responses corresponding to every query vector with probability at least $1 - 1/\mathrm{poly}(n)$.*

*Proof.* Again, we can take a union bound over all $O\left(k \log^2 n\right)$ query vectors to obtain the result. $\square$

**Lemma 15.** *If we draw $\sqrt{\alpha^\star} \log n$ triplets of query vectors $(\mathbf{v} + \mathbf{r}, (q-1)\mathbf{r}, \mathbf{v} + q\mathbf{r})$ randomly as described, then at least one of the triplets is good with probability at least $1 - 1/n$.*

*Proof.* Now, the probability of a triplet of vectors $(\mathbf{v} + \mathbf{r}, (q-1)\mathbf{r}, \mathbf{r})$ being not good is less than $1 - \frac{1}{\sqrt{\alpha^\star}}$ and therefore the probability of all the $\sqrt{\alpha^\star} \log n$ triplets being not good is less than

$$\left(1 - \frac{1}{\sqrt{\alpha^\star}}\right)^{\log n \sqrt{\alpha^\star}} \leq e^{-\log n} \leq n^{-1}$$

which proves the statement of the lemma. □

**Lemma 16.** *For a good triplet of vectors* $(\mathbf{v} + \mathbf{r}, (q-1)\mathbf{r}, \mathbf{v} + q\mathbf{r})$, *we can obtain* $\langle \mathbf{v}, \beta^i \rangle$ *for all* $i \in [L]$.

*Proof.* Recall that since we queried the vector $\mathbf{v} + q\mathbf{r}$, we can simply check which element (say $x$) from the set $\{\langle \mathbf{v} + \mathbf{r}, \beta^i \rangle\}_{i=1}^{L}$ and which element (say $y$) from the set $\{\langle (q-1)\mathbf{r}, \beta^i \rangle\}_{i=1}^{L}$ adds up to an element in $\{\langle \mathbf{v} + q\mathbf{r}, \beta^i \rangle\}_{i=1}^{L}$. It must happen that the elements $x$ and $y$ must correspond to the same unknown vector $\beta^i$ for some $i \in [L]$ because the triplet of vectors $(\mathbf{v} + \mathbf{r}, (q-1)\mathbf{r}, q\mathbf{r})$ is good. Hence computing $x - (y/(q-1))$ allows us to obtain $\langle \mathbf{v}, \beta^i \rangle$ and this step can be done for all $i \in [L]$. □

**Algorithm 3 (Alignment step):** Let a particular good query vector triplet be $(\mathbf{v}^\star + \mathbf{r}^\star, (q^\star - 1)\mathbf{r}^\star, \mathbf{v}^\star + q^\star \mathbf{r}^\star)$. From now, we will consider the $L$ elements $\{\langle \mathbf{r}^\star, \beta^i \rangle\}_{i=1}^{L}$ (necessarily distinct) to be labels and for a vector $\mathbf{u}$, we will associate a label with every element in $\{\langle \mathbf{u}, \beta^i \rangle\}_{i=1}^{L}$. The labelling is correct if, for all $i \in [L]$, the element labelled as $\langle \mathbf{r}^\star, \beta^i \rangle$ also corresponds to the same unknown vector $\beta^i$. Notice that we can label the elements $\{\langle \mathbf{v}^\star, \beta^i \rangle\}_{i=1}^{L}$ correctly because the triplet $(\mathbf{v}^\star + \mathbf{r}^\star, (q^\star - 1)r^\star, \mathbf{v}^\star + q^\star \mathbf{r}^\star)$ is good and by applying the reasoning in Lemma 16. Consider another good query vector triplet $(\mathbf{v}' + \mathbf{r}', (q' - 1)\mathbf{r}', \mathbf{v}' + q'\mathbf{r}')$ which we will call *matching good* with respect to $(\mathbf{v}^\star + \mathbf{r}^\star, (q^\star - 1)r^\star, \mathbf{v}^\star + q^\star \mathbf{r}^\star)$ if it is good and additionally, the vector triplet $(\mathbf{r}', \mathbf{r}^\star, \mathbf{r}' + \mathbf{r}^\star)$ is also good.

**Lemma 17.** *For a fixed known good query vector triplet* $(\mathbf{v}^\star + \mathbf{r}^\star, (q^\star - 1)r^\star, \mathbf{v}^\star + q^\star \mathbf{r}^\star)$, *the probability that any randomly drawn query vector triplet* $(\mathbf{v}' + \mathbf{r}', (q - 1)\mathbf{r}', \mathbf{v}' + q'\mathbf{r}')$ *is matching good with respect to* $(\mathbf{v}^\star + \mathbf{r}^\star, (q^\star - 1)r^\star, \mathbf{v}^\star + q^\star \mathbf{r}^\star)$ *is at least* $\frac{1}{\sqrt{\alpha^\star}}$.

*Proof.* From Lemma 12, we know that the probability that a randomly drawn query vector triplet $(\mathbf{v}' + \mathbf{r}', (q - 1)\mathbf{r}', \mathbf{v}' + q'\mathbf{r}')$ is not good is at most $L^3 \left(\frac{2}{4z^\star + 1} - \frac{1}{(4z^\star + 1)^2}\right)$. Again, we must have for a fixed triplet of indices $i, j, k \in [L]$ such that they are not identical

$$\Pr(\langle \mathbf{r}', \beta^i \rangle + \langle \mathbf{r}^\star, \beta^j \rangle = \langle \mathbf{r}' + \mathbf{r}^\star, \beta^k \rangle)$$
$$= \Pr(\langle \mathbf{r}', \beta^i - \beta^k \rangle = \langle \mathbf{r}^\star, \beta^k - \beta^j \rangle) \leq \frac{1}{4z^\star + 1}$$

Taking a union bound over all non-identical triplets (at most $L^3$ of them), we get that

$$\Pr((\mathbf{r}', \mathbf{r}^\star, \mathbf{r}' + \mathbf{r}^\star \text{ is not good}) \leq \frac{L^3}{4z^\star + 1}$$

Taking a union bound over both the failure events, we get that

$$\Pr((\mathbf{v}' + \mathbf{r}', (q - 1)\mathbf{r}', \mathbf{v}' + q'\mathbf{r}') \text{ is not matching good})$$
$$\leq L^3 \left(\frac{3}{4z^\star + 1} - \frac{1}{(4z^\star + 1)^2}\right)$$
$$\leq 1 - \frac{1}{\sqrt{\alpha^\star}}$$

which proves the lemma. □

**Lemma 18.** *For a matching good query vector triplet* $(\mathbf{v}' + \mathbf{r}', (q - 1)\mathbf{r}', \mathbf{v}' + q\mathbf{r}')$, *we can label the elements in* $\{\langle \mathbf{v}', \beta^i \rangle\}_{i=1}^{L}$ *correctly by querying the vector* $\mathbf{r}' + \mathbf{r}^\star$.

*Proof.* Since $(\mathbf{v}' + \mathbf{r}', (q-1)\mathbf{r}', \mathbf{v}' + q\mathbf{r}')$ is good and we have also queried $\mathbf{v}' + q\mathbf{r}'$, we can partition the set of elements $\{\langle \mathbf{v}' + \mathbf{r}', \beta^i \rangle\}_{i=1}^L \cup \{\langle (q-1)\mathbf{r}', \beta^i \rangle\}_{i=1}^L$ into groups of two elements each such that the elements in each group correspond to the same unknown vector $\beta^i$ as in the reasoning presented in proof of Lemma 16. Again, since $(\mathbf{r}', \mathbf{r}^\star, \mathbf{r}' + \mathbf{r}^\star)$ is good and we have queried $\mathbf{r}' + \mathbf{r}^\star$, we can create a similar partition of the set of elements $\{\langle \mathbf{r}', \beta^i \rangle\}_{i=1}^L \cup \{\langle \mathbf{r}^\star, \beta^i \rangle\}_{i=1}^L$ and multiply every element by a factor of $q - 1$. For each of the two partitions described above we can align two groups together (one from each partition) if both groups contain $\langle (q-1)\mathbf{r}', \beta^i \rangle$ for the same $i \in L$ (the values $\langle \mathbf{r}', \beta^i \rangle$ are necessarily distinct and therefore this is possible). Hence, for every $i \in [L]$, we can compute $\langle \mathbf{v}', \beta^i \rangle$ correctly and also label it correctly because of the alignment. $\qquad\square$

**Algorithm 3 (Putting it all together)** First, we condition on the event that for all batches of queries (number of batches will be polynomial in $k$ and $\log n$) we make, the denoised means are extracted correctly which happens with probability at least $1 - \frac{1}{n}$ by Corollary 14. As described in Algorithm 3, in the first step we sample a pair of vectors $(\mathbf{v}, \mathbf{r})$ such that $\mathbf{v}$ is uniformly drawn from $\{-1, +1\}^n$ and $\mathbf{r}$ is uniformly drawn from $\{-2z^\star, -2z^\star + 1, \ldots, 2z^\star - 1, 2z^\star\}^n$. We also sample a random number $q$ uniformly and independently from the set $\{1, 2, \ldots, 4z^\star + 1\}$ and subsequently, we use batches of queries of size $c_4 L^2 \log n \exp((\sigma/\epsilon)^{2/3})$ corresponding to the three vectors $\mathbf{v} + \mathbf{r}, (q-1)\mathbf{r}$ and $\mathbf{v} + q\mathbf{r}$ respectively. We will repeat this step for $\sqrt{\alpha^\star} \log n + c'\alpha^\star k \log(n/k)$ iterations. Additionally, for each query vector pair $((\mathbf{v_1}, \mathbf{r_1})$ among the first $\sqrt{\alpha^\star} \log n$ iterations and for each vector pair $((\mathbf{v_2}, \mathbf{r_2})$ among the latter $c'\alpha^\star k \log(n/k)$ iterations, we also make the batch of queries corresponding to the vector $\mathbf{r_1} + \mathbf{r_2}$. From Lemma 15, we know that with probability at least $1 - \frac{1}{n}$, one of the query vector triplets among the first $\sqrt{\alpha^\star} \log n$ triplets is good. Moreover, it is also easy to check if a query vector triplet is good or not and therefore it is easy to identify one. Once a good query vector triplet $(\mathbf{v}^\star + \mathbf{r}^\star, (q^\star - 1)\mathbf{r}^\star, \mathbf{v}^\star + q^\star\mathbf{r}^\star)$ is identified, it is also possible to correctly identify matching good query vectors among the latter $c'\alpha^\star k \log(n/k)$ query vector triplets with respect to the good vector triplet. We now have the following lemma characterizing the number of matching good query vector triplets:

**Lemma 19.** *The number of matching good query vector triplets from $\alpha^\star c' k \log(n/k)$ randomly chosen triplets is at least $c' k \log(n/k)$ with probability at least $1 - \left(\frac{k}{n}\right)^{\tilde{c}k}$ for some constant $\tilde{c} > 0$.*

*Proof.* For a randomly drawn query vector triplet, we know that it is matching good with probability at least $\frac{1}{\sqrt{\alpha^\star}}$ from Lemma 17. Since there are $\alpha^\star c' k \log(n/k)$ query vector triplets drawn at random independently, the expected number of matching-good triplets is at least $\sqrt{\alpha^\star} c' k \log(n/k)$. Further, by using Chernoff bound [5], we can show that

$$\Pr(\text{Number of matching good triplets} < c' k \log(n/k))$$

$$= \Pr\left(\text{Number of matching good triplets} < \sqrt{\alpha^\star} c' k \log(n/k) \left(1 - \frac{\sqrt{\alpha^\star} - 1}{\sqrt{\alpha^\star}}\right)\right)$$

$$\leq \exp\left(-\frac{(\sqrt{\alpha^\star} - 1)^2 c' k \log(n/k)}{2\sqrt{\alpha^\star}}\right).$$

$\qquad\square$

From Lemma 18, we know that for every matching good query vector triplet $(\mathbf{v}' + \mathbf{r}', (q-1)\mathbf{r}', \mathbf{v}' + q\mathbf{r}')$, we can label the elements in $\{\langle \mathbf{v}', \beta^i \rangle\}_{i=1}^L$ correctly and from Lemma 19, we know that we have aggregated de-noised query measurements corresponding to $c' k \log(n/k)$ vectors randomly sampled from $\{+1, -1\}^n$. However, since we have specifically picked $c' k \log(n/k)$ matching good vectors after the entire scheme, we do not know which query vectors will be matching good apriori and therefore we need to have the following guarantee:

**Lemma 20.** *From $\alpha^\star c' k \log(n/k)$ vectors randomly chosen from $\{+1, -1\}^n$, any $c' k \log(n/k)$ vectors scaled by a factor of $1/\sqrt{c' k \log(n/k)}$ will satisfy the $\delta -$ RIP property with high probability.*

The proof of this lemma is delegated to the appendix. Now we are ready to proof the main theorem in this setting.

*Proof of Theorem 5.* The total number of batches of queries made is at most $3c'\alpha^\star k \log(n/k) \log n$. Further, recall that size of each batch that is sufficient to recover the denoised means accurately is $c_4 \log n \log(\sigma/\epsilon)^{2/3}$. Hence the total number of queries is $O\left(k(\log n)^3 \exp(\sigma/\epsilon)^{2/3}\right)$ as mentioned in the theorem statement. From Lemma 18 and Lemma 20, we know that for every vector $\{\beta^i\}_{i=1}^L$, we have $c'k \log(n/k)$ linear query measurements such that the measurement matrix scaled by $1/\sqrt{c'k \log(n/k)}$ has the $\delta - \mathsf{RIP}$ property. Therefore, it is possible to obtain the best $k$-sparse approximation of all the vectors $\beta^1, \beta^2, \ldots, \beta^L$ by using efficient algorithms such as Basis Pursuit. $\qquad\square$

Now Theorem 3 follows as a corollary.

*Proof of Theorem 3 for general $L$.* Notice that the query with the largest magnitude of query response that we will make is $\mathbf{v} + (4z^\star + 1)\mathbf{r}$ where $\mathbf{v}$ is sampled from $\{+1, -1\}^n$ and $\mathbf{r}$ is sampled from $\{-2z^\star, -2z^\star + 1, \ldots, 2z^\star - 1, 2z^\star\}$. Therefore, we must have

$$\mathbb{E}|\langle \mathbf{v} + (4z^\star + 1)\mathbf{r}, \beta^i\rangle|^2$$
$$= \mathbb{E}|\langle \mathbf{v}, \beta^i\rangle|^2 + (4z^\star + 1)^2 \mathbb{E}|\langle \mathbf{r}, \beta^i\rangle|^2$$
$$= 1 + (4z^\star + 1)\sum_{i=1}^{2z^\star} i^2$$
$$= 1 + \frac{z^\star(2z^\star + 1)(4z^\star + 1)^2}{3}.$$

since $||\beta^i||_2 = 1$. Since the variance of the noise $\mathbb{E}\eta^2$ is $\sigma^2$, we must have that

$$\mathsf{SNR} = \frac{1}{\sigma^2}\left(1 + \frac{z^\star(2z^\star + 1)(4z^\star + 1)^2}{3}\right).$$

Substituting the above expression in the statement of Theorem 5 and using the fact that $z^\star$ is a constant, we get the statement of the corollary. $\qquad\square$

# E   Proof of Lemma 20

First, let us introduce a few notations. For a given any set of indices $\mathbf{T} \subset [n]$, denote by $\mathbf{X_T}$ the set of all vectors in $\mathbb{R}^n$ that are zero outside of $T$. We start by stating the Johnson-Linderstrauss Lemma proved in [2].

**Lemma 21.** *[Lemma 5.1 in [2]] Let $\mathbf{A}$ be a $m \times n$ matrix such that every element in $\mathbf{A}$ is sampled independently and uniformly at random from $\{1/\sqrt{m}, -1/\sqrt{m}\}$. For any set $T \subset [n]$ such that $|T| = k$ and any $0 < \delta < 1$, we have*

$$(1 - \delta)||\mathbf{x}||_2 \le ||\mathbf{Ax}||_2 \le (1 + \delta)||\mathbf{x}||_2 \quad \text{for all } \mathbf{x} \in \mathbf{X_T}$$

*with probability at least $1 - 2(12/\delta)^k e^{-\frac{m}{2}(\delta^2/8 - \delta^3/24)}$.*

We are now ready to prove Lemma 20. Since there are $\binom{n}{k}$ distinct subsets of $[n]$ that are of size $k$, we take a union bound over all the subsets and therefore the failure probability of Lemma 21 for all sets of indices of size $k$ (definition of $\delta$-RIP) is at most

$$2(12/\delta)^k \binom{n}{k} e^{-\frac{m}{2}(\delta^2/8 - \delta^3/24)}.$$

We need that from $\alpha m (\alpha > 1)$ vectors randomly sampled from $\{\frac{1}{\sqrt{m}}, \frac{-1}{\sqrt{m}}\}^n$ any $m$ vectors satisfy the $\delta$-RIP property for some value of $m$. Therefore, the probability of failure is at most

$$2\left(\frac{12}{\delta}\right)^k \binom{n}{k}\binom{\alpha m}{m} e^{-\frac{m}{2}(\delta^2/8 - \delta^3/24)}.$$

By Stirling's approximation and the fact that both $\alpha m$ and $m$ is large, we get that

$$\binom{\alpha m}{m} \approx \sqrt{\frac{\alpha}{2\pi m(\alpha - 1)}} \left(\frac{\alpha^\alpha}{(\alpha-1)^{\alpha-1}}\right)^m$$

Further we can also upper bound the binomial coefficients $\binom{n}{k}$ by $\left(\frac{en}{k}\right)^k$. Hence we can upper bound the failure probability as

$$\exp\left(-m(\delta^2/16 - \delta^3/48) + m\log\left(\frac{\alpha^\alpha}{(\alpha-1)^{\alpha-1}}\right) + k\log(en/k) + \log(12/\delta) + \log 2\right)$$

Therefore, if we substitute $m = c'k\log(en/k)$ for some constant $c' > 0$, we must have the failure probability to be upper bounded as $e^{-c''m(1+o(1))}$ for some $c'' > 0$ as long as we have

$$c'\left(\frac{\delta^2}{16} - \frac{\delta^3}{48}\right) > c'\log\left(\frac{\alpha^\alpha}{(\alpha-1)^{\alpha-1}}\right) + 1$$

implying that

$$\frac{\alpha^\alpha}{(\alpha-1)^{\alpha-1}} < \exp\left(\frac{\delta^2}{16} - \frac{\delta^3}{48} - \frac{1}{c'}\right).$$

Hence, by choosing the constant $c'$ appropriately large, the term in the exponent on the right hand side can be made positive. Since the left hand side of the equation is always greater than $1$, there will exist an $\alpha$ satisfying the equation.