[Reviews · NeurIPS 2019]

Reviewer 1



Originality: The paper relies on techniques similar to "Sampling Signals With Finite Rate of Innovation" by Martin Vetterli, Pina Marziliano and Thierry Blu as well as "Sketching for Large-Scale Learning of Mixture Models" by Nicolas Keriven, Anthony Bourrier, Rémi Gribonval, Patrick Pérez. This type of algorithms should be mentioned in the state of the art part. The Finite Rate of Innovation theory highly relies on the so called "annihilating filters" form error correction coding, which is highly relevant for the submitted publication. The submitted paper tries to remove the assumption on common support of sparse vectors. Quality: The paper is well written. Clarity: Structure of the paper is clear. Due to lack of experiments and conclusion, the reader might have a feeling as if a paragraph or a page is missing. Significance: The problem at hand is of great interest for the scientific community and has a broad range of applications.

Reviewer 2



1. The dependence of SNR is extreme. It scales as an exponential function. I wonder whether it only occurs in the proof or a fundamental limitation of the approach. The authors did not provide a empirical comparison to any competing method even to [27] on which the presented algorithm improves. It would be interesting to see how the algorithm competes with the state-of-the-art in its empirical performance particularly in the presence of noise. 2. The statement of Theorem 3 hid the dependence on L by assuming L = O(1) outside the theorem. Isn't the proof providing any dependence on L? 3. Some key definitions are missing. For example, \epsilon-precision is not clearly stated when it first appears in line 194. The authors can add just a single equation for the definition. 4. In Algorithm 1, it has not been clearly stated how a k-sparse vector is recovered from 2k Vandermonde measurements. It suffices for the identifiability. But the recovery by a practical algorithm requires more measurements. While algorithm and how many more measurements suffices for its guarantee need to be clarified. 5. In Algorithm 1, step 6, how does one choose L unique results out of L\log(Lk^2)? 6. The proof of Theorem 1 looks interesting but unfortunately I was not able to fully understand it. Some critical steps need more explanation. Line 154: does the interval here need to belong the set of positive real number? Can it include any negative numbers? How does draw the consistency in Line 155 from the above line. This step does not look obvious to me. I think the consistency is the key to draw the main conclusion but I lost the track at this point. Hope the authors can elaborate on their proof. 7. The min-distance estimator in Line 15 of Algorithm 2 is one of the main ingredient and needs to be included in the main text not in the appendix. 8. The paragraph after Lemma 2 is not very clear. In line 225, the authors say that the estimation of means in the scalar Gaussian mixture can be obtained up to a finite precision. But anyway \beta_i's are assumed to be of a finite precision. In the next line, the authors assumed the knowledge of exact means without any error. I am not following the flow of logic in this paragraph. In the statement of Algorithm 2, in many locations it is assumed that we have access to error-free estimates of oracle information, which are inner products of v_i and \beta_j. It would help if the authors can confirm that these exact estimates available by a practical algorithm by Lemma 2. 9. Finally, continued from #1, it would be interesting to see (even without a comparison to any other method), how sensitive the algorithm is to additive noise. I am curious about the empirical performance of a component that estimates scalar means of a Gaussian mixture in SampleAndRecovery per varying SNR.

Reviewer 3



Summary: This paper considers the problem of recovering multiple sparse vectors from their linear (possibly noisy) measurements, under the setting that one can query a vector and the oracle returns the measurement computed at one of the unknown sparse vectors. Both noiseless case and noisy case are considered, and in each case the authors propose an algorithm and analyze its query complexity. The proposed algorithm has three major steps: first, query the same vector multiple times and separate the measurements corresponding to different sparse vectors; second, identify the measurements of different query vectors with the same sparse vector; third, recover the sparse vectors using the convectional compressed sensing recovery algorithm. Strength: The proposed algorithms seems new and has stronger theoretical guarantees (e.g., weaker condition in a more general setting) than the previous work. Weakness: - I am concerned about the practical performance of the proposed algorithms. Specifically, Algorithm 1 uses Vandermonde matrix, which is known to be very ill-conditioned and numerically unstable, especially for large $n$ and small $\alpha$. Since $\alpha$ can be as small as $1/k^2L^2$, I am worried that the algorithm may fail in practice even for moderate size $n$, $k$, and $L$. The paper is more convincing if the authors could provide experimental results for the proposed algorithms. - Section 3.2 (which is claimed to be the main contribution of this paper) skips a lot of the technical details, making it hard for readers to fully understand Algorithm 3. For example, what is the intuition behind the three constants defined in Line 259-261; how do we know when a triplet is good or not (Line 268-270) in practice; the alignment step (Line 274-280) is difficult to follow. I would suggest moving the pseudocode to the appendix, and explain the algorithm in more details in the paper. - No time and space complexity is provided for the proposed algorithm. Minor: In the abstract the author briefly mention the connection between their algorithm and error correcting codes. It would be better to describe that connection in more detail when presenting the algorithm. -----------------After reading the rebuttal------------------- Thanks for the clarification on distinguishing between good or bad triplets. I understand that this work is mainly theoretical and would still appreciate it if small empirical results (or even toy examples) can be added to help evaluate the algorithm in practice. Section 3.2 (which is a major contribution of this paper) skips a lot of the technical details. It would be better if the authors can explain the denoising and alignment steps in more details. For example, move Lemma 18 from the appendix to the main paper so that the readers can understand how Algorithm 3 matches two good query triplets. Overall Section 2 and 3.1 are well-written. The theoretical results are new and interesting. I will increase my score.

[Author Response · NeurIPS 2019]

We would like to thank the reviewers for their insightful comments. Responses below:

**Common Points**:

• We want to emphasize that our work is primarily theoretical, building on [27] (published in IEEE Transaction
on Information Theory) by improving sample complexity bounds and more importantly eliminating the need
for strong assumptions. We believe this significantly enhances our understanding of this fundamental problem.

• Given this, we feel that experiments do not add much to the paper. Synthetic experiments will simply confirm
our theorems. We urge the reviewers to evaluate the paper on the basis of the theoretical results and techniques.

• We also note that many published NeurIPS do not contain empirical evaluations, and this is not a barrier to
publication or even NeurIPS awards. For example one of last year's best paper awards went to *Nearly Tight*
*Sample Complexity Bounds for Learning Mixtures of Gaussians via Sample Compression Schemes*, a purely
theoretical contribution, on a problem quite close to what we study here.

**Reviewer 1:**

• Thanks for pointing out the relevant papers. The second paper is quite relevant as it studies mixture model
parameter recovery in a compressive sensing framework. We will add the citations.

• An illustration is a great suggestion! We will add one to the final version to help explain our algorithms.

**Reviewer 2:**

• Exponential SNR is common in statistical problems involving mixtures. Examples include trace reconstruction
and learning binomial mixture where the dependence is known to be optimal. The exponential dependence
in our result arises from a reduction to learning Gaussian mixtures. This latter problem is extremely well
studied (and out of scope for our paper), and while we are not aware of better results for Gaussian mixtures,
any improvement would immediately yield improvements for our setting.

• As in [27], we assume $L$ is a constant. We can extract the dependence on $L$ from our proof for the final version,
but note that it is fairly complicated as $L$ is implicit in the definitions of the constants in Eqs (A), (B), (C).

• $\epsilon$-precision means that all coordinates are integer-multiples of $\epsilon$ (this assumption is also used in [27]). We can
add the definition to the camera ready.

• The set of responses recovered after using 2k rows of the Vandermonde matrix is provably unique, and it is
possible to recover the unknown vector uniquely using efficient decoding algorithms borrowing from literature
of coding theory (such as, Berlekamp-Welch Decoder, see, Arora and Barak, Computational Complexity,
Sec. 19.3).

• There are at most $L$ unique responses (fixed sensing vector, only $L$ possible choices for $\beta$), and by a coupon
collector phenomenon $L \log(Lk^2)$ measurements suffices to ensure there are exactly $L$, with high probability.

• The interval must belong to the set of positive real numbers so that we can apply Lemma 1. If the difference
polynomials have no roots in an interval, then by continuity, the ordering of the polynomials $f^1, f^2, \ldots, f^L$
does not change in that interval. Thus the ordering is consistent.

• We will add the definition of the minimum distance estimator to the final version.

• We will clarify the paragraph after Lemma 2. In short, the $\beta^j$s are on an $\epsilon$-grid, but $\langle v, \beta^j \rangle$ may not be if we
naively run the noiseless algorithm. (The $\epsilon$-grid property is crucial for our mixture learning algorithm.) This
motivates the new decoding strategy, which ensures $\langle v, \beta^j \rangle$ is on the $\epsilon$-grid.

**Reviewer 3:**

• You are right that Vandermonde matrices are ill-conditioned – this is precisely why we do not use them for the
noisy case. That is also the point of having different treatments for noisyless and noisy cases, with the later
being more involved. This is similar to classical results in compressed sensing.

• Due to space restrictions we cannot include the entire proof in the body. But all the details are in the appendix.

• The constants are simply artifacts of the proof.

• Note that every triple (good or bad) contains vectors with integral entries. Thus we can learn the parameters
$\langle v, \beta^j \rangle$ for the three vectors. Then to check if the condition holds, note that for fixed $\beta$ and our choice of
vectors, we have $\langle v_1, \beta \rangle + \langle v_2, \beta \rangle = \langle v_3, \beta \rangle$, so we can match up the learned parameters. If we find two such
identities involving a single parameter, we know that triplet is not good.

[Meta-Review · NeurIPS 2019]

Overall the solid theoretical results in this paper were enough to push it over the bar, despite a few complaints of lack of experiments. The reviewers were in consensus to accept the paper.